# Over-Expression of Dehydroascorbate Reductase Improves Salt Tolerance, Environmental Adaptability and Productivity in *Oryza sativa*

**DOI:** 10.3390/antiox11061077

**Published:** 2022-05-28

**Authors:** Young-Saeng Kim, Seong-Im Park, Jin-Ju Kim, Sun-Young Shin, Sang-Soo Kwak, Choon-Hwan Lee, Hyang-Mi Park, Yul-Ho Kim, Il-Sup Kim, Ho-Sung Yoon

**Affiliations:** 1Research Institute of Ulleung-do & Dok-do, Kyungpook National University, Daegu 41566, Korea; kyslhh1228@hanmail.net; 2School of Life Sciences, BK21 Plus KNU Creative BioResearch Group, Kyungpook National University, Daegu 41566, Korea; sheep91528@naver.com (S.-I.P.); deenamon@naver.com (J.-J.K.); 3Department of Biology, College of Natural Sciences, Kyungpook National University, Daegu 41566, Korea; snowmooa@hanmail.net; 4Plant Systems Engineering Research Center, Korea Research Institute of Bioscience and Biotechnology, Daejeon 34141, Korea; sskwak@kribb.re.kr; 5Department of Molecular Biology, Pusan National University, Busan 46241, Korea; chlee@pusan.ac.kr; 6National Institute of Crop Science, Rural Development Administration, Jeonju 54875, Korea; parkhm2002@korea.kr; 7Highland Agriculture Research Institute, National Institute of Crop Science, Rural Development Administration, Pyeongchang 25342, Korea; kimyuh77@korea.kr; 8Advanced Bio-Resource Research Center, Kyungpook National University, Daegu 41566, Korea

**Keywords:** ascorbate, ascorbate-glutathione system, dehydroascorbate reductase, reactive oxygen species, yield parameters

## Abstract

Abiotic stress induces reactive oxygen species (ROS) generation in plants, and high ROS levels can cause partial or severe oxidative damage to cellular components that regulate the redox status. Here, we developed salt-tolerant transgenic rice plants that overexpressed the dehydroascorbate reductase gene (*OsDHAR1*) under the control of a stress-inducible sweet potato promoter (*SWPA2*). *OsDHAR1*-expressing transgenic plants exhibited improved environmental adaptability compared to wild-type plants, owing to enhanced ascorbate levels, redox homeostasis, photosynthetic ability, and membrane stability through cross-activation of ascorbate-glutathione cycle enzymes under paddy-field conditions, which enhanced various agronomic traits, including root development, panicle number, spikelet number per panicle, and total grain yield. *dhar2*-knockdown plants were susceptible to salt stress, and owing to poor seed maturation, exhibited reduced biomass (root growth) and grain yield under paddy field conditions. Microarray revealed that transgenic plants highly expressed genes associated with cell growth, plant growth, leaf senescence, root development, ROS and heavy metal detoxification systems, lipid metabolism, isoflavone and ascorbate recycling, and photosynthesis. We identified the genetic source of functional genomics-based molecular breeding in crop plants and provided new insights into the physiological processes underlying environmental adaptability, which will enable improvement of stress tolerance and crop species productivity in response to climate change.

## 1. Introduction

Rice is a major staple crop worldwide [1], and although rice yield is compromised by land erosion, degradation, and climate change, the demand for rice production is steadily increasing. Of late, the climate change caused by global warming has intensified. The variables that represent the most serious threats to the environment have been debated upon by environmentalists, policy makers, farmers, and scientists. This problem has become an environmental, political, economic, scientific, and global issue [2,3,4]. At the plant and field scales, climate change is likely to crosstalk with rising CO_2_ concentrations and other environmental factors, such as high salinity, nutrient depletion, drought, flooding, extreme temperatures, UV radiation, heavy metal contamination, and photochemical cycles, thereby affecting the physiological processes of crop plants and reducing agricultural productivity [5,6]. 

In the AsA-GSH system, AsA peroxidase (APX) catalyzes the reduction of H_2_O_2_ to H_2_O using AsA and NAD(P)H. During this process, AsA is oxidized to monodehydroascorbate (MDHA) and further oxidized to dehydroascorbate (DHA) through spontaneous oxidation. AsA is subsequently regenerated from DHA and MDHA by the action of MDHA reductase (MDHAR) and DHA reductase (DHAR), respectively [7,8,9]. DHAR is a critical enzyme for AsA pool maintenance and redox homeostasis because unstable DHA molecules are spontaneously and irreversibly hydrolyzed to 2,3-diketogulonic acid if they are not rapidly reduced to AsA [10]. Therefore, DHA must be recycled to AsA by DHAR, using GSH as a substrate. DHAR has been cloned and characterized from a variety of plant species, including spinach [11], rice [12], *Arabidopsis thaliana* [13], and sweet potato [14]; high *DHAR* expression has been reported to enhance tolerance toward abiotic stress. For example, the overexpression of wheat [15], human [16], and *Arabidopsis* [17,18] *DHAR* genes has been shown to improve tolerance toward ozone, chilling, drought, and salt in transgenic tobacco, and the overexpression of the rice *DHAR* has been shown to increase the tolerance of transgenic *Arabidopsis* [19] and *Escherichia coli* [20] to salt and oxidative stress. These results suggest that *DHAR* is an important factor in the detoxification of abiotic stress-induced reactive oxygen species (ROS).

Although several studies have examined the *DHAR*-mediated abiotic stress response of plants grown under controlled conditions, the findings of these studies are unlikely to correspond to the stress responses of plants under natural conditions for several reasons [21]. First, in controlled experiments, specific stressors are generally applied suddenly or transiently for a specific time, which does not correspond to the stressors experienced by plants grown under natural conditions where plants are constantly exposed to stressors. Second, only the effects of one or two stressors are generally investigated in controlled experiments, even though plants are often simultaneously exposed to multiple stressors under natural conditions. Third, controlled conditions impede the elucidation of the mechanisms underlying environmental adaptability because they generally fail to compensate for seasonal variations that occur under such natural conditions [21,22]. Indeed, in rice, the investigation of plants grown under natural conditions can also be problematic because leaf phenology and environmental stress can influence metabolic processes and alter circadian rhythms; however, long-term investigations of the physiological responses of rice plants to their environment are necessary to cope with climate change [21,22]. However, to date, the role of *DHAR* in mediating the environmental adaptability and productivity of rice grown under natural paddy fields has not been examined.

Given the complexity associated with studying environmental stress responses in plants, in the present study, we aimed to elucidate the relationship between *OsDHAR1*-mediated AsA recycling and ROS-induced oxidative stress that results from environmental stress in paddy fields. Recently, we identified the AsA-regeneration mechanism (i.e., the ping-pong mechanism) by determining the protein structure of thermostable OsDHAR1 [23]. The mechanism underlying the molecular response of higher plants to environmental stresses has been analyzed by studying a number of genes responding to various stresses at the transcript level [24]. We postulated that *OsDHAR1* overexpression would increase the environmental adaptability and productivity of rice plants grown in paddy fields in response to multiple environmental stresses. It is important to analyze the function of stress-inducible genes, not only for further understanding of the mechanisms underlying stress tolerance and responses of higher plants, but also for improving the stress tolerance of crops by gene manipulation.

Here, we used gene expression profiling and genetic network to analyze the expression patterns in *OsDHAR1*-overexpressing plants under conditions of environmental stress, and to identify the target genes of stress-related transcription factors and potential cis-acting DNA elements by combining the expression data. The profiles of plant responses to environmental stresses are expected to enable the identification of regulators that will be useful in biotechnological approaches aimed at improving stress tolerance, in addition to providing new insights into the physiological processes involved in the mechanisms underlying environmental adaptability.

## 2. Materials and Methods

### 2.1. Vector Construction and Agrobacterium-Mediated Rice Transformation

Full-length cDNA encoding *Oryza sativa* L. *japonica* cytosolic dehydroascorbate reductase (*OsDHAR1*) (accession no. AY074786) was cloned from rice leaves using reverse-transcription polymerase chain reaction (RT-PCR). The primers, OsDHAR-FC and OsDHAR-RC were used for PCR (Appendix A). The cDNA—designated as *OsDHAR1*—was subsequently cloned into the *Nco*I/*Kpn*I site of the pBT1 vector, downstream of the *SWPA2* promoter. The *SWPA2*::*OsDHAR1*::*Tnos* terminator was cloned into the *Hind*III site of pCAMBIA3300 (Figure 1A) and subsequently designated as pOsDHAR1. The promoter and *OsDHAR1* were sequenced to confirm that the open reading frame was intact, i.e., there were no frame shifts or nucleotide conversions. Finally, the pOsDHAR1 binary vector was introduced into *Agrobacterium* strain LBA 4404, which was then used to transform rice calli from the scutellum of the mature seeds of *O*. *sativa* L*. japonica* ‘Ilmi variety,’ as described previously [25,26].

### 2.2. Plant Materials and Growing Conditions

The Ilmi variety of *O*. *sativa* L*. japonica* was used as the host for the homologous overexpression of *OsDHAR1*. Approximately 20 independent T_0_ transgenic rice plants were screened using Basta. Among the transformants, 13 independent T_1_ transgenic plants grown from T_1_ seeds were selected through Basta and PCR genotyping. Four independents homozygous T_2_ plants were identified using the same process, and one independent transgenic line overlapped with another independent transgenic line. Three independent transgenic lines (TR1, TR2, and TR3) were selected for subsequent experiments. For genotyping, both transgenic and wild-type (WT) plants were cultivated in natural paddy fields located at the Gunwi campus (36°24 N, 128°53 E, 95 m a.s.l.) of Kyungpook National University, Korea, from June to October. Seeds were germinated at 32 °C in a controlled growth chamber, transplanted into soil pots (12.5 cm diameter), and maintained at 28–32 °C under a 16-h light/8-h dark cycle. For the salt stress assay, 4-week-old seedlings were treated with 100 mM NaCl for 25 days. Leaves were harvested when seedlings were exposed to 100 mM NaCl for 4 days or grown for ∼4 weeks after transplantation in paddy fields. The harvested leaves were immediately soaked in liquid nitrogen and stored at −70 °C until subsequent use. In addition, a T-DNA insertion mutant line (3A-03259), a putative dehydroascorbate reductase gene (*OsDHAR2*, Os06g0232600) knockdown mutant, was provided by the Salk Institute Genomic Analysis Laboratory (http://signal.salk.edu/cgi-bin/RiceGE, 18 November 2019). Homozygous lines were identified as described for the *DHAR1*-overexpressing transgenic plants.

### 2.3. Copy Number and Location of the OsDHAR1 Transgene

Determination of the copy number and location of the *OsDHAR1* transgene consisted of two steps. First, genomic DNA was digested with restriction enzymes and ligated to an adaptor, and second, the fragments were subjected to PCR amplification using two primer sets that are specific to the *OsDHAR1* transgene and adaptor regions. In the present study, genomic DNA (500 ng) was digested with 2 U of *Msp*I and *Hinc*II and ligated using 5 U of T4 DNA ligase (Takara Bio) in 20-micoliter reactions; the reaction mixtures contained T4 DNA ligase buffer and 50 pmol of adaptors and were incubated for 1 h at 37 °C. The first PCR was conducted in 20-microliter reactions that contained 0.5 pmol of each primer (ADA1 with either LB1 or RB1, and ADA2 with either LB2 or RB2) (Appendix A) and 1 μL digestion/ligation product, with the following conditions: an initial denaturation step at 95 °C for 5 min, 20 cycles of 30 s at 94 °C and 1 min at 72 °C, and a final elongation step at 72 °C for 10 min. A second PCR was conducted with 5 μL of the first PCR product under the following conditions: an initial denaturation step at 94 °C for 5 min; 40 cycles of 30 s at 94 °C, 30 s at 60 °C, and 1 min at 72 °C; and a final elongation step at 72 °C for 10 min. The amplified products were separated on a 1% agarose gel, purified using a HiYield Gel/PCR DNA Extraction Kit (Real Biotech, Taipei, Taiwan), and sequenced using LB2 and RB2 primers (Appendix A).

### 2.4. Measurement of the Ascorbate Content

The total AsA (tAsA), AsA, and DHA contents were spectrophotometrically determined as described previously [27]. Briefly, ∼0.2 g of frozen leaf sample was ground with a mortar and pestle and vortexed vigorously for 10 min after adding 2 mL of 5% (*v*/*v*) *m*-phosphoric acid. The homogenate was centrifuged at 12,000× *g* for 20 min and the clear supernatant was collected. The tAsA content was determined in a reaction mixture containing 100 μL crude extract, 500 μL 150 mM KH_2_PO_4_ buffer (pH 7.4; 5 mM EDTA), and 100 μL 10 mM dithiothreitol, which reduced DHA to AsA. The reaction mixture was incubated at 25 °C for 10 min and supplemented with 100 μL of 0.5% (*w*/*v*) N-ethylmaleimide (NEM) to remove excess dithiothreitol. The AsA content was assayed by replacing dithiothreitol and NEM with 200 μL of deionized H_2_O as described previously. Color was developed in both reaction mixtures by adding 400 μL of 10% (*w*/*v*) trichloroacetic acid, 400 μL of 44% (*v*/*v*) *o*-phosphoric acid, 400 μL of α,α-dipyridyl prepared in 70% (*v*/*v*) ethanol, and 200 μL of 30% FeCl_3_. The reaction mixture was incubated at 40 °C for 1 h and the absorbance was quantified at 525 nm. The DHA concentration was estimated by subtracting the AsA content from the tAsA content.

### 2.5. Enzyme Activity Assay

To prepare the crude protein extracts for the MDHAR and DHAR assays, leaf samples (0.2 g fresh weight) were homogenized at 4 °C in 1 mL of extraction buffer (50 mM Tris-HCl (pH 7.5), 3 mM MgCl_2_, 1 mM EDTA, 1 mM PMSF, and protease inhibitor cocktail (Sigma-Aldrich, St. Louis, MI, USA)) with a mortar and pestle. The homogenate was centrifuged at 15,000× *g* for 20 min at 4 °C and the clear supernatant was used for enzymatic assays. To prepare crude protein extracts for APX and GSH reductase (GR) assays, leaf samples (0.2 g) were extracted in 1 mL of extraction buffer (mM potassium phosphate (pH 7.0), 7 mM β-ME, and EDTA-free protease inhibitor cocktail (Roche, Basel, Switzerland)). In addition, crude protein extract for the APX assay was prepared in the presence of 2 mM AsA. All enzyme assays were performed by monitoring the corresponding reaction mixtures at 25 °C using a UV-1800 UV-VIS spectrophotometer (Shimadzu, Kyoto, Japan). MDHAR activity was assayed in 1 mL reaction mixture containing 50 mM potassium phosphate (pH 7.2), 0.2 mM NADH, 2 mM AsA, 1 U AsA oxidase, and crude extract. The absorbance was measured at 340 nm, the activity was calculated using an absorbance coefficient of 6.2 mM^−1^cm^−1^; and one unit of MDHAR activity was defined as the amount of enzyme that oxidized 1 nmol of NADH per min at 25 °C [28]. DHAR activity was assayed in 1 mL reaction mixture containing 50 mM potassium phosphate (pH 7.0), 0.1 mM EDTA, 0.5 mM DHA, 2.5 mM GSH, and crude extract. The absorbance was measured at 265 nm, the activity was calculated using an absorbance coefficient of 14.6 mM^−1^cm^−1^ [29], and one unit of DHAR activity was defined as the amount of enzyme that produced 1 nmol of AsA per min at 25 °C. GR activity was assayed in 1 mL reaction mixture containing 50 mM potassium phosphate (pH 7.6), 1.7 mM EDTA, 1.0 mM GSSG, 0.1 mM NADPH, and crude extract. Absorbance was measured at 340 nm [30]. APX activity was assayed in 1 mL reaction mixtures containing 50 mM sodium phosphate (pH 7.0), 0.2 mM EDTA, 0.5 mM AsA, 0.2 mM EDTA, 2.5 mM H_2_O_2_, and crude extract (50 μg equivalent). Enzyme activity was recorded as the decrease in absorbance at 290 nm for 1 min, and the amount of oxidized AsA was calculated using an extinction coefficient of 2.8 mM^−1^ cm^−1^ [31]. All enzymatic activities in the transgenic plants were calculated relative to those in the WT plants grown under normal conditions; enzyme activities observed in the WT plants were set as 100%. 

### 2.6. Redox State Analysis

Leaf samples (0.2 g fresh weight) were homogenized at 4 °C in 1 mL extraction buffer (20 mM sodium phosphate (pH 7.2), 1 mM PMSF, and EDTA-free protease inhibitor cocktail (Roche)), using a mortar and pestle. The homogenate was centrifuged at 15,000× *g* for 20 min at 4 °C and the clear supernatant was collected. Intracellular hydroperoxide levels were determined based on ferrous ion oxidation in the presence of xylenol orange, a ferric ion indicator. Briefly, crude extract (50 μL) was added to 950 μL of FOX reagent (100 μM xylenol orange, 250 μM ammonium ferrous sulfate, 100 mM sorbitol, and 25 mM sulfuric acid). The mixture was incubated at 25 °C for 30 min and then centrifuged to remove the flocculated material before measuring the absorbance at 560 nm [32]. Lipid peroxidation was spectrophotometrically determined by measuring malondialdehyde or thiobarbituric acid-reactive substances, as reported previously [33]. Briefly, 1.0 mL crude extract was mixed with 2.0 mL of TCA-thiobarbituric acid-HCl reagent (15% thiobarbituric acid, 0.375% TCA, and 0.25 N HCl), mixed thoroughly, and heated in a boiling water bath (95 °C) for 15 min. After cooling, the supernatant was centrifuged at 12,000× *g* for 10 min, and the absorbance of the sample was measured at 535 nm and compared with that of a control sample. Malondialdehyde concentration in the sample was calculated using an extinction coefficient of 1.56 × 10^5^ M^−1^ cm^−1^ [34]. For 3,3′-diaminobenzidine staining, the leaves of whole rice plants were sprayed with 50 µM methyl viologen solutions containing 0.1% Tween-20 and incubated for 12 h, after which the leaves were cut, placed in 3,3′-diaminobenzidine (1 mg mL^−1^)-HCl (pH 3.8), incubated for 8 h at 28 °C, and then cleared in boiling ethanol (96%) for 10 min before photographing [35].

### 2.7. Ion Leakage and Chlorophyll Fluorescence Analysis under Conditions of MV Treatment

Five leaves were collected from the TR and WT rice plants grown for 4 weeks in the natural paddy field, after which ten leaf discs were cut (approximately 1 cm). The leaf discs were then incubated in water supplemented with 10 μM methyl viologen (MV) in the dark at 25 °C for 12 h to allow the MV to diffuse into the leaf discs. After pre-incubation, leaf discs were placed in a growth chamber (16 h light: 8 h dark) at 25 °C for 120 h. The conductivity of the suspension was measured using a conductance meter (455C; Isteck Co., Ltd., Seoul, Korea) before and after autoclaving at 121 °C for 15 min to release all electrolytes. Relative ion leakage was expressed as a percentage of total conductivity [36]. MV-treated leaves of TR and WT plants were dark-adapted for 15 min. Chlorophyll fluorescence was measured based on the photosynthetic activity of photochemical yield (*Fv/Fm*), which represents the maximum yield of the photochemical reaction of photosystem II at room temperature, using a Handy Fluor Cam FC 1000-H (Photon Systems Instruments Ltd., Brno, Czech Republic) [37]. *Fv*/*Fm* values were measured as described previously [38].

### 2.8. Microarray-Based Gene Expression Profiling

Leaf samples (pooled from five plants per family) were harvested from WT and TR1 plants that had been grown for ∼30 days under natural paddy field conditions. TR1 plants were used as representative transgenic lines. Total RNA was extracted from rice leaves using the TRIzol reagent (Sigma-Aldrich) according to the manufacturer’s instructions. Following homogenization, 1 mL of the solution was transferred to a 1.5 mL Eppendorf tube and centrifuged at 12,000× *g* for 10 min at 4 °C to remove any insoluble material. The supernatant containing the RNA was collected, mixed with 0.2 mL of chloroform, and centrifuged at 12,000× *g* for 15 min at 4 °C. The upper phase containing the RNA, was transferred to a new tube, and RNA was precipitated by mixing the solution with isopropyl alcohol (0.5 mL) and recovered by centrifuging the sample at 12,000× *g* for 10 min at 4 °C. The RNA pellet was washed briefly with 1 mL of 75% ethanol by centrifuging at 7500× *g* for 5 min at 4 °C. Finally, the RNA pellet was dissolved in DNase-free water and RNA quality and quantity were assessed using an Agilent Bioanalyzer 2100 (Agilent Technologies, Santa Clara, CA, USA). Gene expression was analyzed using GeneChip Rice Genome Arrays (4 × 44 K; Affymetrix), which (each) comprised over 54,000 probe sets, and 11 pairs of oligonucleotide probes were synthesized in situ on the arrays. Biotinylated cRNA was prepared from 100 ng of total RNA, according to the standard Affymetrix protocol. Following fragmentation, 15 µg of RNA was hybridized for 16 h at 45 °C using a GeneChip Rice Genome Array. GeneChips were washed and stained in the Affymetrix Fluidics Station 450 and scanned using the Affymetrix GeneChip Scanner 3000 7G. The data were analyzed using robust multi-array analysis—with default Affymetrix settings—together with the global scaling normalization method. The trimmed mean target intensity of each array was arbitrarily set at 100, and both the normalized and log-transformed intensity values were analyzed using GeneSpring GX 13.0 (Agilent Technologies). The fold-change filters used defined the up- and downregulated genes as those that are expressed at levels that were at least 200% less than 50% of the expression of the corresponding genes in the controls. Hierarchical clustering was performed using GeneSpring GX 13.0 (Agilent Technologies), with the clustering algorithm being the Euclidean distance and average linkage. The genetic network of the differentially expressed genes was framed using String (http://apps.cytoscape.org/apps/stringapp, 18 December 2020) and ClueGo (http://apps.cytoscape.org/apps/cluego, 18 December 2020).

### 2.9. Estimation of Agronomic Traits in Paddy Fields

To examine the yield parameters of the transgenic rice plants, three independent T_3_ or T_4_ homozygous lines and their respective controls (WT plants) were transplanted into natural paddy fields located at the Kyungpook National University campus in 2020 (T_3_) and 2021 (T_4_). A completely randomized block design was employed with two replicates, and each plot comprised 12 plants, with two rows for each line, 0.2 m between plants, and 0.3 m between plots. When the plants reached maturity and the grains ripened, yield components were scored by digging up 8 rice plants per plot to a radius of 15 cm. After washing the plant roots, total plant weight (g), stem weight (g), and root weight were measured and number of panicles per hill and number of spikelets per panicle were counted. The filling rate (%) of the rice plants was calculated relative to 100% of WT rice plants, which were defined had expressed the degree of the filling rate as panicles and spikelets under field conditions. To measure the total grain weight (g), and 1000-grain weight (g), whole seeds per plant were collected and weighed, and then 1000 seeds were counted and weighed additionally. Unfilled and filled grains were separated, randomly counted, and weighed. The following agronomic traits were scored: total plant weight (g), culm weight (g), root weight (g), number of panicles per hill, number of spikelets per panicle, filling rate (%), total grain weight (g), and 1000-grain weight (g).

### 2.10. Data Analysis

All experiments were performed with at least three independent repetitions, and the results are expressed as mean ± standard deviation (SD). All biochemical results of the transgenic plants were calculated relative to those of WT plants grown under normal conditions, which were defined as 100%.

## 3. Results

### 3.1. Development of SWPA2::OsDHAR1 Transgenic Rice Plants

To assess the stress tolerance of *OsDHAR1*-overexpressing plants, we produced 20 lines (T_0_ generation) of transgenic plants carrying the *SWPA2*::*OsDHAR1* transgene construct. Fifteen seeds (T_1_ seeds) per primary transgenic line were cultivated in a paddy field and their progenies (T_2_ seeds) were harvested. 

In the present study, three independent salt-tolerant homozygous transgenic families (TR1, TR2, and TR3) were identified (Figure 1B). In each of these families, only a single *SWPA2*::*OsDHAR1* transgene copy was inserted into the rice genome (Figure 1C; Appendix A). In addition, all salt-tolerant plants harbored the *SWPA2*::*OsDHAR1* transgene, which was stably inherited by the following generation. 

To investigate the mechanism by which *OsDHAR1* overexpression increases salt tolerance, the phenotypes and gene expression of the transgenic plants were examined under salt stress. There was no difference in the phenotypes of the transgenic and WT plants grown under normal conditions (0 day; Figure 1D). Under conditions of salt stress, most of the WT plants withered after 25 days, but the transgenic plants were relatively unaffected (25 days; Figure 1D). The survival rate of the transgenic plants was higher than that of the WT plants. In addition, the transgenic and WT plants exhibited clear phenotypic differences with respect to their ability to recover from salt damage. More specifically, the transgenic plants recuperated quickly after exposure to salt stress, whereas the WT plants did not (Figure 1D), and the fresh weight of the transgenic plants was approximately 2-fold greater than that of the WT plants (Figure 1G). We constructed *dhar2*-knockdown transgenic lines (*dhar2-1* and *dhar2-2*) and further analyzed the response of the mutant to salt stress. As a result, the *dhar2*-knockdown mutants showed hypersensitivity to salt stress (Appendix A). Homologous overexpression of *OsDHAR1* was confirmed by measuring the levels of *OsDHAR*1 mRNA and protein in 4-week-old seedlings exposed to 100 mM NaCl for 4 days. The analysis revealed that *OsDHAR1* was expressed at greater levels in the transgenic plants than in the WT plants (Figure 1E), and western blotting confirmed that the salt-stressed transgenic plants expressed higher levels of the OsDHAR1 protein (Figure 1F). On the other hand, the *dhar2*-knockdown mutants showed weak *DHAR* expression compared to in WT plants (Appendix A). Thus, our results confirmed that the expression of *OsDHAR1* is related to stress, and the *OsDHAR1* transgene is effectively translated into a functional DHAR protein, as expected. 

### 3.2. Performance of OsDHAR1-Expressing Transgenic Plants under Salt Stress

We investigated, at least partially, the *OsDHAR1*-mediated mechanism underlying the enhanced salt tolerance of transgenic plants by challenging healthy plants with salt stress and then analyzing the biochemistry of the leaf samples. In the following experiments, TR1 and TR2 plants, which were better than TR3 plants in Figure 1, were used as representatives of three independent transgenic plants. The activity of MDHAR, APX, and GR, which are involved in the AsA-GSH cycle, was greatly increased by salt stress in the transgenic plants (TR1 and TR2) (compared to WT plants), even though the WT plants themselves exhibited a moderate increase in enzyme activity. Further, APX and GR exhibited the highest and lowest activities, respectively, in transgenic plants (Figure 2A). Under normal conditions, the enzyme activity of the transgenic plants was slightly elevated, possibly because of transgene insertion. 

To evaluate the metabolic effect of *OsDHAR1* overexpression, the AsA pool of leaves from transgenic (TR1 and TR2) and WT plants was compared. Although salt stress increased the AsA levels in both transgenic and WT plants, the AsA levels of the transgenic plants were higher than those of the WT plants. Meanwhile, the DHA level of the transgenic plants was distinctly decreased compared to that of the WT plants, which subsequently resulted in an increased AsA/DHA ratio. Indeed, under conditions of salt stress, the AsA/DHA ratio of the transgenic plants was ∼1.67-fold higher than that of the WT plants, whereas, only a slight difference was observed under normal conditions (Figure 2B). These results indicated that the DHAR activity of the *OsDHAR1*-expressing transgenic plants is increased, and the *dhar2* mutants, in contrast, showed reduced DHAR activity under salt stress conditions (Appendix A). Based on these results, hydroperoxide and malondialdehyde levels were measured to determine whether the increased AsA pool of the transgenic plants affected the redox state. Under salt stress, the hydroperoxide and malondialdehyde contents of the transgenic plants were approximately 1.5-fold lower than those of the WT plants (Figure 2C), and there was no difference under normal conditions. Ion leakage of the transgenic plants was also lower than that of the WT plants under conditions of salt stress (Figure 2D). Therefore, our results indicate that *OsDHAR1* overexpression improves redox homeostasis, the AsA pool, and membrane permeability in transgenic rice plants by cross-activating antioxidant enzymes, including MDHAR, APX, GR, and DHAR.

### 3.3. Agronomic Traits of OsDHAR1-Expressing Transgenic Plants

We postulated that *OsDHAR1* overexpression could protect transgenic plants from other types of environmental stresses because the *OsDHAR1*-transgenic plants exhibited greater salt tolerance owing to their improved AsA pool and redox state. To test this hypothesis, we transplanted three independent T_3_ or T_4_ homozygous transgenic (TR1–TR3) plants and WT plants into paddy fields using a completely randomized design. The plants were grown to the seed maturation stage, and agronomic parameters were scored from the vegetative to the reproductive stage. Transgenic plants performed better than the WT plants during both the vegetative and post-flowering harvest stages (Figure 3B and Appendix A). These observations prompted us to examine other yield components. In 2020 and 2021, the total plant weight, culm weight, root weight, number of panicles per hill, number of spikelets per panicle, filling rate, total grain weight, and 1000-grain weight were 20.7 and 19.6%, 17.4 and 17.9%, 12.7 and 12.3%, 17.2 and 20.3%, 14.9 and 16.8%, 11.0 and 9.6%, 18.2 and 20.4%, and 3.7 and 5.0% greater, respectively, in transgenic plants than in WT plants (Figure 3C). The rice grain yield factors TGW of the transgenic plants was higher than that of the WT plants and appeared to result from an increased number of spikelets per panicle and 1000-grain weight of the transgenic plants. Therefore, the number of panicles per hill in transgenic plants increased by ∼18.7% compared to that in WT plants, and as a result, the total grain weight of transgenic plants increased by ∼19.3%. There was also no difference in the agronomic traits of the transgenic plants grown over the two years, despite obvious differences in the mean temperature, precipitation, humidity, and total sunshine (Figure 3A). On the contrary, the field-grown *dhar2* mutant plants exhibited reduced agronomic traits (Appendix A). More specifically, the total plant weight, culm weight, root weight, number of panicles per hill, number of spikelets per panicle, filling rate, total grain weight, and thousand seeds weight of the *dhar2* mutant plants was ∼48.0, 30.8, 71.6, 19.3, 17.5, 35.0, 51.3, and 44.6% lower, respectively, than those of the WT plants (Appendix A). Among these traits, the reduced total biomass (total plant weight and root weight) and poor seed maturation of *dhar2* mutant plants distinctly diminished rice grain yield (total grain weight and hundred seeds weight). These results indicated that *OsDHAR1* overexpression significantly enhanced the rice grain yield and total biomass in response to various environmental stresses under paddy-field conditions.

### 3.4. Effect of OsDHAR1 Overexpression on Overall Gene Expression

As *OsDHAR1*-expressing transgenic plants exhibited increased grain yield and biomass under paddy-field conditions, we performed gene expression profiling to identify the mechanisms underlying *OsDHAR1*-mediated environmental tolerance. TR1 plants were used because in this line, the *SWPA2*::*OsDHAR1* transgene was inserted into an intergenic region. We found that ∼404 genes were upregulated, and ∼525 genes were downregulated (Figure 4; Appendix A). The upregulated genes were related to amino acid (glycine, serine, and threonine) metabolism, glycosphingolipid biosynthesis, porphyrin and chlorophyll metabolism, cell growth, transport, and translation; genes related to lipid metabolism were highly expressed (Appendix A). In contrast (Appendix A), the downregulated genes were related to nitrogen metabolism, amino acid (phenylalanine, tyrosine, and tryptophan) biosynthesis, organic metabolism, organic biosynthesis, cellular metabolism, and cellular biosynthesis. 

There was also a distinct difference in the expression of genes involved in biological regulation and biosynthetic processes in WT and TR1 plants. Briefly, the genes, Os03g0115800, Os11g0206000, and Os03g0115800 encoding conserved hypothetical proteins were most highly expressed, the expression of *OsDHAR1* was increased by 2.4-fold, and the expression of the transcription factors *HSfA2a, HSfA2d*, *TAGL 12*, *AT103*, *WRKY10*, and *Os_F0727* was upregulated in the TR1 plants. The following were the upregulated gene categories: pathogenesis-related protein, planthopper resistance, isoflavone reductase homolog IRL, heat shock proteins and molecular chaperonins, root-specific protein (*RCc2*), plant secretory proteins, rapid alkalization factor 2, tubulin (*OsTUB6*), actin depolymerizing factor (*OsADF3*), annexin p35, photosystems, phytochrome-interacting protein, molybdopterin biosynthesis CNX2 protein, sodium/hydrogen exchanger protein, abiotic stress responsive protein (*OsASR2*), heavy metal transport/detoxification protein, heat- and acid-stable phosphoprotein, drought-induced protein, ABA/WDS induced protein, multidrug-resistance associated protein, jasmonate ZIM-domain protein (*OsJAZ2*), methyl chloride transferase which is harmless to ozone layer 1 (*OsHOL1*), aquaporin, glutaredoxins (*OsGRX4* and *OsGRX29*), peroxidase (*Prx59*), MDHAR (*OsMDHAR1*), AsA peroxidases (*OsAPX1* and *OsAPX8*), GR1, and AsA biosynthesis (L-galactono-1,4-lactone dehydrogenase) (Appendix A; Appendix A). The highly expressed genes were networked into categories that corresponded to the regulation of signal transduction, glycosphingolipid biosynthesis, hydrolase activity, protein complex subunit organization, zinc ion transmembrane transporter activity, chlorophyll envelope, response to ROS, water-soluble vitamin biosynthesis, redox homeostasis, and isoprenoid biosynthesis (Figure 5 and Figure 6; Appendix A). 

Meanwhile, the genes encoding RNA-directed DNA polymerase, F-box domain-containing proteins, and ankyrin repeat domain-containing proteins exhibited the greatest downregulation in the TR1 plants; the following gene categories were downregulated: transcription factors (*OsMYB42/85*, *OsMYB45*, *Spl7*, *OsNAC103*, *OsbZIP49*, *OsWRKY1*, *OsWRKY10*, *OsWRKY69*, and *CBF-B*), nuclear Y/CCAAT-box binding factor A (*OsHAP2E*), zinc fingers (*OsPUB67*, *OsRFHC-7*, and *OsBBX23*), TonB box, APETALA2/ethylene-responsive protein 129, ethylene responsive element binding factor (*OsERF3*), symbiosis-related disease resistance protein, multi-antimicrobial extrusion protein, apoptosis regulator Bcl-2 protein (*OsBAG1*), ATPase, physical impedance induced protein, alcohol dehydrogenase, lactate dehydrogenase, cadmium tolerant 1, flavonoid 3,5-hydroxylase 2, AsA peroxidase (*OsAPX7*), glutaredoxins (*OsGRX2* and *OsGRX16*), ferredoxin III, GSH transferases (*OsGST4*, *OsGST7*, *OsGST23*, and *OsGST30*), auxin transport protein (*REH1*), and auxin-up RNA26 (*OsSAUR26*; Appendix A). The downregulated genes were networked to categories that corresponded to the MAPK signaling pathway, (secondary active) transmembrane transporter activity, O-methyltransferase activity, aromatic amino acid family biosynthesis, and phenylalanine ammonia-lyase activity (Appendix A). Notably, *OsDHAR1* was linked to *OsGSTU42*, *GalDH*, *OsAPX1*, *OsGSTU7*, and *OsGSTU23* (Appendix A).

A complete genetic network of the differentially expressed genes, generated using ClueGO (Appendix A) and String (Appendix A), indicated that *OsDHAR*-mediated environmental adaptability was precisely regulated between the genes, despite the network being highly complex. Taken together, our results suggest that *OsDHAR1* overexpression improves the environmental adaptability of TR1 plants grown in paddy fields by co-activating a wide range of cell rescue systems, including various transcription factors, the AsA-GSH cycle, aquaporins, heavy metal detoxification, ion exchange, cytoskeleton, photosynthesis, isoprenoid and isoflavone biosynthesis, pathogenesis-related proteins, and several unknown conserved hypothetical proteins. 

## 4. Discussion

High salinity, which inhibits plant growth and photosynthesis in crop plants, is one of the key factors that are most detrimental to rice productivity, especially as soil salinization increases every year owing to anthropgenic activities and climate change. Approximately 6.5% of the world’s total area and ∼20% of the irrigated lands have been severely salinized [39,40]. Therefore, there is a need to improve salt tolerance in crop plants. 

In the present study, *OsDHAR1*-overexpressing transgenic rice plants exhibited improved salt tolerance by enhancing their AsA pool, redox homeostasis, membrane stability, and ion homeostasis following ion leakage. Transgenic plants achieve this through co-activation of enzymes (MDHAR, APX, and GR) involved in the AsA-GSH cycle (Figure 1 and Figure 2). The physiological and molecular mechanisms of salt tolerance in various plant species have been widely reported [41], and ‘omics’-based plant molecular breeding has been widely employed to increase crop tolerance to salt stress [39,40]. Rice plants also exhibit various adaptive mechanisms to overcome salt damage, including regulation of seedling vigor, and efficient transport of salt ions through the root system [39]. For example, the specific overexpression of ion transporters and pump-responsive genes that encode Na^+^/H^+^ antiporter (*SOS1*, *SOD2*, *nhaA*, and *NHX*), Na^+^ transporter (*HKT1*), Na^+^/K^+^ antiporter (*HKT2*), Na^+^ ATPase (*ENA*), and vacuolar pyrophosphatase (*vacuolar H^+^-PPase*) has been shown to improve salt tolerance in a variety of transgenic crop plants by increasing biomass (shoot and root) production, germination rate, and proline content by enhancing ion homeostasis [41]. Taken together, our findings suggest that *OsDHAR1* plays an important role in salt tolerance by improving the AsA pool and redox and ion homeostasis, indicating that *DHAR* expression in WT plants is insufficient to overcome salt stress.

The transgenic plants exhibited a better phenotype than the WT plants under salt stress (Figure 1), which was due to the improvement in redox homeostasis and the AsA pool through the cross-activation of MDHAR, APX, and GR (Figure 2). We hypothesized that high salinity induces ROS-induced oxidative stress. Salt-induced oxidative stress has been recognized as an important source of cellular damage in plants [42], and ROS scavenging depends on both enzymatic and non-enzymatic systems. Among these systems, the most important non-enzymatic antioxidant is AsA, which functions as a reductant for peroxidases [43,44]. In the present study, the greater AsA levels in the transgenic plants resulted from increased MDHAR and DHAR activities (Figure 2A,B). Using AsA as an electron donor, APX catalyzes the reduction of H_2_O_2_, which enhances redox homeostasis and proteostasis by minimizing oxidative damage through the buffering of stress-induced ROS [44]. During exposure to salt stress, GR activity also differed between transgenic and WT plants. The enzyme activity was ∼1.3-fold higher in the transgenic plants than in the WT plants (Figure 2A), indicating that MDHAR and GR were more efficient in converting oxidized AsA (MDHA or DHA) to reduce AsA. 

In contrast, the *dhar2*-knockdown transgenic lines, *dhar2-1* and *dhar2-2*, which exhibited lower DHAR activity, were sensitive to salt stress (Appendix A). According to previous reports, homologous overexpression of *DHAR* (CrDHAR1; Cre10.g456750) enhances the growth and survival of *Chlamydomonas reinhardtii* through the improvement of redox state, photosynthesis efficiency, GSH and AsA pools, and GR and APX activities, which subsequently improve resistance to ROS-induced oxidative stress [45]. Previous studies have shown that increasing AsA concentrations through the overexpression of AsA biosynthesis genes enhances stress tolerance in transgenic tobacco, *Arabidopsis,* and lettuce [46], and the overexpression of ROS detoxification-responsive genes, namely *APX*, *GST*, *SOD*, *MDHAR*, and *CAT,* increases salt tolerance in transgenic tobacco and rice by augmenting photosynthetic efficiency and plant growth [41]. In contrast, several *Arabidopsis DHAR*-knockout mutants (*dhar1*, *2*, and *3*) that exhibited decreased tAsA and increased DHA were more sensitive to bright light and oxidative stress than the WT plants. The mutants possessed lower levels of reduced AsA, MDHAR, and DHAR activities and higher levels of hydroperoxide, which indicates that AsA can limit ROS scavenging under abiotic stress [44,47]. *CrDHAR1*-knockdown microRNA lines with reduced DHAR expression and AsA recycling ability were sensitive to light and oxidative stress [45]. 

The salt-tolerant barnyard grass *Echinochloa crus-galli* L. depends on higher activities of guaiacol peroxidase, APX, and GR than WT plants when exposed to 200 mM NaCl, whereas salt-sensitive plants with reduced enzyme activities exhibited slower growth rates, higher lipid peroxidation rates, and higher levels of ROS in their leaves [42]. Moreover, in salt-stressed *Vicia faba* seedlings, exogenous supplementation of AsA was reported to alleviate oxidative stress [48,49], which indicates that steady-state AsA content was insufficient for effective antioxidative defense under salt stress and that its biosynthesis and recycling were necessary [42]. Taken together, the results of the present study suggest that enhanced AsA recycling via elevated DHAR activity and the combination of enzymes involved in the AsA-GSH pathway increases salt tolerance and that *OsDHAR1* can be used to facilitate the development of improved rice cultivars with enhanced AsA content.

During the vegetative to reproductive stages of growth and development, rice plants grow in paddy fields under a variety of environmental conditions, including temperature, drought, flooding, ozone, UV radiation, nutrient depletion, air pollution, and photoperiod, which can decrease productivity [50]. To overcome the factors that limit rice productivity, researchers have concentrated on improving the yield of superior phenotypes using up- and downstream regulatory genes [51,52]. Among these genes, the expression levels of *OsDHAR1*, which are a downstream regulatory gene, are higher in the three independent homologous TR rice lines than in WT rice plants in paddy fields at transcriptional and translational levels, as a result, *OsDHAR1* overexpressing TR plants have been shown to affect environmental adaptability and productivity in rice grown in natural paddy fields, which are frequently exposed to abiotic and biotic stressors (Appendix A). In the present study, transgenic rice plants overexpressing *OsDHAR1* exhibited better phenotypes than WT plants in both the vegetative (Appendix A) and reproductive (Figure 3B) stages. The major physiological differences resulted from the improvement of redox state, photosynthetic ability, and membrane permeability, which was achieved by buffering ROS-induced oxidative damage through the cross-activation of MDHAR, APX, GR, and DHAR in transgenic plants (Appendix A). The increased tAsA in the transgenic plants resulted from the induction of the AsA biosynthesis gene *GalDH*. Augmented MDHAR and DHAR activities increased the AsA/DHA ratio by elevating and reducing AsA and DHA levels, respectively, in transgenic plants, which then enhanced the AsA pool via AsA recycling (Appendix A). Plants that are susceptible to abiotic and biotic stresses are generally depleted of AsA. In contrast, stress-resistant plants exhibit increased AsA levels and a steady-state or decreased oxidation status, depending on whether DHA is increased along with AsA or remains constant [21,53], as in the present study. Therefore, changes in the AsA pool following the AsA/DHA ratio can be used as an indicator of stress response ability in the field.

In the present study, field-grown transgenic rice plants that overexpressed *OsDHAR1* possessed significantly improved agronomic traits, including rice grain yield and total biomass (root and shoot), compared to WT plants (Figure 3). The roots and shoots of the transgenic plants were longer and healthier than those of the WT plants, which contributed to enhanced total biomass. The total plant weight, culm weight, root weight, number of panicles per hill, number of spikelets per panicle, filling rate, total grain weight, and 1000-grain weight of the transgenic plants were ∼20.1, 17.6, 12.5, 18.7, 15.3, 10.3, 19.3, and 4.3% greater, respectively, than those of the WT plants during 2020 and 2021, even though the environmental conditions varied significantly between the two years (Figure 3C). In particular, the increase in grain yield was caused by increased panicle architecture (number of panicles per hill), spikelet density (number of spikelets per panicle), and 1000-grain weight-based total grain weight (Figure 3C). In contrast, field-grown salt-sensitive *dhar2* plants exhibited poor phenotypes during both the vegetative and reproductive stages, and as a result, the *dhar2* mutants exhibited reduced agronomic traits (Appendix A). Specifically, the total plant weight, culm weight, root weight, number of panicles per hill, number of spikelets per panicle, filling rate, total grain weight, and 1000-grain weight of the *dhar2* plants were ∼44.0, 28.4, 72.7, 20.1, 19.0, 36.7, 57.5, and 48.4% lower, respectively, than those of the WT plants (Appendix A). Among these traits, the reduced total biomass (total plant weight and root weight) and poor seed maturation of *dhar2* mutants distinctly diminished rice grain yield (total grain weight and 1000-grain weight). Indeed, AsA content is correlated with seed development and seedling growth [46]. AsA is present in its reduced state during early embryonic development but undergoes progressive oxidation so that DHA is more prevalent by the time the seeds reach maturity [46,54,55]. Thus, DHA produced during seed germination must be rapidly recycled into AsA.

In previous studies, field-grown transgenic rice plants that overexpress cell rescue-related genes have exhibited improved growth, development, and yield. More specifically, plants that overexpressed *OsLSK1* (large spike S-domain receptor-like kinase 1), *OsLRK1* (leucine-rich repeat receptor-like kinase 1), *OsEBS1* (enhancing biomass and spikelet number), *AtSUC2* (*A. thaliana* phloem-specific sucrose transporter), and *CaPLA1* (Capsicum annuum phospholipiase A1) have been reported to possess more tillers, larger spikelet hulls, longer panicles, more branches per panicle, heavier grains, and greater cellular proliferation than WT plants, thereby improving total grain yield [56,57,58,59]. Meanwhile, the overexpression of *OsAGSW1* (ABC1-like kinase related to grain size and weight) has been shown to significantly increase grain size, grain weight, grain filling rate, and 1000-grain weight by regulating the number of external parenchyma cells and the development of vascular bundles, which results in wider and longer spikelet hulls [60]. In addition, elevated expression of *OsNRT2.3b* (nitrate transporter), *OsSPL14* (squamosal promoter binding protein-like 14), *OsAMT* (ammonia transporter), and *OsPTR9* (peptide transporter/nitrate transporter 1 homologue) in transgenic rice enhances pH homeostasis by increasing the uptake of ions (N, Fe, and P), improving grain yield, and improving nitrogen use efficiency [61,62,63,64]. In contrast, the repression of *OsCKX2*, which encodes cytokinin oxidase 2, increases tiller number and rice yield, whereas constitutive overexpression reduces tiller number and growth in a gene dose-dependent manner [65]. Taken together, it is apparent that genes involved in signaling pathways and transport drive changes in yield-associated traits, whereas antioxidant genes such as *OsDHAR1* do not. In addition, our findings also suggest that *OsDHAR1* affects functional tolerance to environmental stresses and consequently leads to improvement in growth, seed maturation (shape and size), and productivity of rice plants grown in paddy fields. 

Plant growth depends on the synergistic interactions of genes or proteins, and because the yield potential of crop plants is determined by the complex interactions of these factors, we investigated the expression profiles and network maps of a large number of genes in paddy fields [22]. Microarray-based gene expression profiling revealed that *OsDHAR1* mediates the environmental adaptability of transgenic rice plants grown under paddy field conditions, despite the underlying mechanism being complex. In the *OsDHAR1*-expressing TR1 plants, genes involved in a wide range of functional categories, including cell cycle, transcription factors, amino acid (serine and threonine) metabolism, glycosphingolipid biosynthesis, photosynthesis process, secretory proteins, cell rescue systems (redox and ion homeostasis, and proteostasis), antioxidants (isoflavone and AsA biosynthesis), and metal detoxification, were upregulated. The elevated genes were networked to the gene ontology corresponding to the regulation of signal transduction, glycosphingolipid biosynthesis, hydrolase activity, protein complex subunit organization, membrane transporter activity, chlorophyll envelope, response to ROS, water-soluble vitamin biosynthesis, redox homeostasis, and isoprenoid biosynthesis (Figure 5 and Figure 6; Appendix A). Genes interacting with *OsDHAR1*, *OsGSTU42*, *GalDH*, and *OsAPX1* were highly expressed in TR1 plants (Appendix A). 

In addition, a wide range of genes encoding heat shock proteins, molecular chaperone proteins and their cofactors, and genes related to AsA biosynthesis and regeneration were also upregulated in TR1 plants (Appendix A). AsA, which is the most abundant antioxidant in a variety of plant species, plays a major role in regulating redox status along with other antioxidants [43]. In addition, AsA also stimulates cell division during embryo development by increasing the proportion of cells progressing through the G1-to-S transition, whereas DHA blocks cell cycle progression and regulates the cell cycle and flowering time [66,67,68]. Accordingly, the suppression of DHAR expression in transgenic tobacco reduces the rates of leaf expansion, delays flowering time, and reduces foliar dry weight [46,54]. High expression of chaperonin-related genes can increase proteostasis by minimizing ROS-induced oxidative damage, such as protein carbonylation, which inhibits and inactivates enzymes, and by enhancing the refolding of aggregated proteins or folding of synthesized proteins under environmental conditions. Moreover, AsA, which functions as an essential cofactor for collagen synthesis enzymes, is linked to AsA-dependent protein folding. High concentrations of DHA enhance the formation of disulfide bonds (oxidative protein folding) in secretory proteins. DHAR harboring protein disulfide isomerase or chaperone-like activity can enhance protein refolding and DHA reduction under oxidative condition [69]. 

On the other hand, the expression of genes involved in nitrogen metabolism, amino acid (phenylalanine, tyrosine, and tryptophan) biosynthesis, apoptotic process, cell adhesion, flavonoid biosynthesis, and auxin-metabolic pathway were downregulated in TR1 plants. The downregulated genes were networked to the gene ontology corresponding to the MAPK signaling pathway, (secondary active) membrane transporter activity, methyltransferase activity, aromatic amino acid family biosynthetic process, and phenylalanine ammonia-lyase activity (Appendix A). Interestingly, the *OsDHAR1*-overexpressing TR1 plants regulated auxin-related genes, which could lead to rice grain yield via panicle and spikelet development, and improved leaf architecture. Accordingly, *BG1* (*big grain 1*)-expressing transgenic rice plants exhibited significantly improved plant biomass, seed weight, and grain yield via regulation of auxin transport. However, the knockdown of *BG1* decreased auxin sensitivity and grain size [70].

The total genetic network indicated that the differentially expressed genes formed a complex network and provided important information about the mechanisms underlying environmental adaptability (Appendix A). Although the mechanism is theoretically explained, it is not known. Recently, gene expression profiling was conducted for specific organs, tissues, and growth stages of field-grown WT rice (*O*. *sativa* L. *japonica*). In Appendix A, this field-based profiling provides baseline information for the functional characterization of genes and reveals critical developmental and physiological transitions involved in growth potential [22]. Taken together, the mutual cooperative network of the differentially expressed genes in *OsDHAR1*-expressing TR1 plants promotes root growth and development, nutrient uptake, ion homeostasis, senescence regulation, pathogen resistance, photosynthesis optimization, redox homeostasis, proteostasis, and lipid metabolism, which together accelerate agronomic performance and productivity by regulating auxin metabolism.

## 5. Conclusions

The present study characterized the functions of *OsDHAR1* with respect to the alleviation of abiotic stress, especially under paddy field conditions. We found that *OsDHAR1*-expressing transgenic plants were more tolerant to environmental stress, including high salinity, and we attributed the *OsDHAR1*-mediated environmental adaptability to increased ROS-neutralizing ability. The observed ROS-neutralizing ability was achieved by enhancing the AsA pool, redox and ion homeostasis, proteostasis, lipid-based membrane stability, photosynthesis capacity, and auxin transport through mutual organic networks between the genes, which improved various agronomic traits, including plant growth (root development), biomass, panicle and tiller number, spikelet per panicle, seed maturation, and grain yield. Furthermore, our findings regarding the molecular, genetic, and phenotypic bases for the responses of rice to abiotic and biotic stresses will facilitate the application of molecular breeding, genetic engineering, and other integrated approaches for developing stress-tolerant crop cultivars.

## Figures and Tables

**Figure 1 antioxidants-11-01077-f001:**
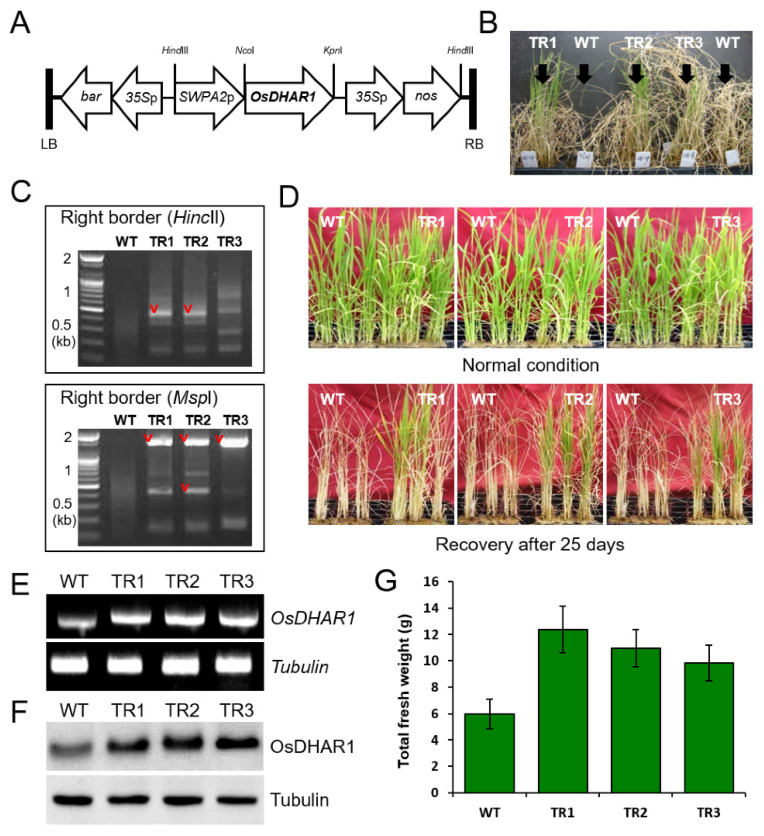
Production and salt stress response of *OsDHAR1*-expressing transgenic rice plants. (**A**) Schematic diagram of the *OsDHAR1* gene construct. *SWPA2*::*OsDHAR1* includes a sweet potato stress-inducible promoter (*SWPA2p*), the *OsDHAR1* coding regions, and the *Tnos* terminator, which was derived from the nopaline synthase gene. The *bar* gene, which encodes the herbicide bialaphos (Basta) and whose expression is driven by the 35S promoter, was used as a selection marker. (**B**) Selection of salt-tolerant homozygous transgenic lines. Three independent transgenic lines (TR1, TR2, and TR3) were identified upon screening 4-week-old seedlings using 100 mM NaCl. (**C**) Visualization of flanking T-DNA sequencing amplicon. Electrophoresis was performed to determine the copy number and insertion position of the integrated *OsDHAR1* transgene. Flanking T-DNA sequencing was performed as described in SI Materials and Methods. The PCR amplicon digested with the indicated restriction enzymes is represented and amplicons marked with arrows were sequenced (date not shown). (**D**) Phenotypes of transgenic and WT seedlings under conditions of salt stress. Upper panel, 4-week-old WT and transgenic plants in normal state; lower panel, 4-week-old WT and transgenic plants recovered after exposure to 100 NaCl for 25 days. Phenotype analysis was performed at least three times and represented as a representative image. (**E**) Confirmation of *OsDHAR1* expression using semi-quantitative RT-PCR and (**F**) western blotting. Signal intensity of WT plants was due to endogenous *DHAR1* expression. *tubulin* (*Tub*) and corresponding protein were used as loading controls. (**G**) Total fresh weight of recovered plants after exposure to 100 NaCl for 25 days. WT, wild-type rice; TR, transgenic rice.

**Figure 2 antioxidants-11-01077-f002:**
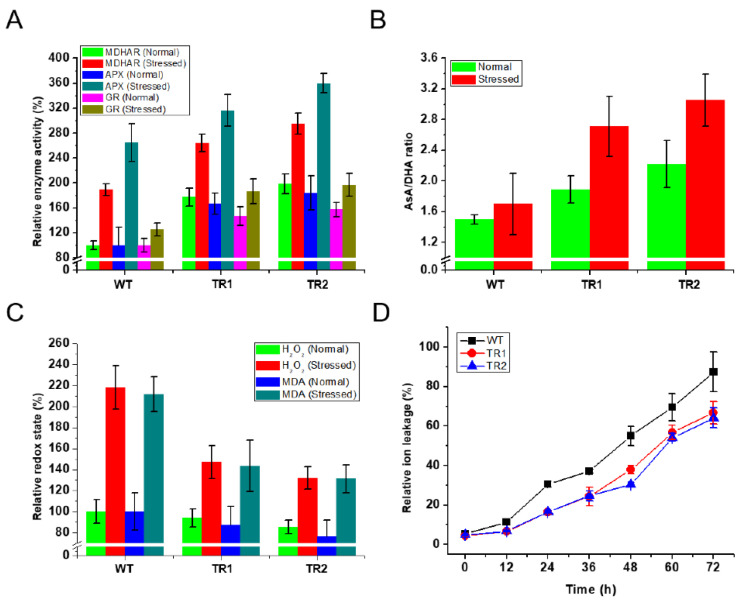
Ascorbate pool and redox homeostasis of transgenic and wild-type rice plants under conditions of salt stress. (**A**) Four-week-old seedlings were exposed to 100 mM NaCl for 4 days at 28–32 °C, and then leaf tissues were harvested for evaluating enzymatic activity, (**B**) AsA/DHA ratio, (**C**) hydroperoxide level and (**D**) ion leakage analyses. Except for the results of the ion leakage analyses, data are expressed as relative percentages of the levels observed in WT plants and represent the means ± SD of at least three replicates from three independent experiments. Ion leakage was calculated by adjusting ion release at the initial time (0 h) in both the WT and transgenic plants. WT, wild-type rice; TR, transgenic rice; AsA, ascorbate; DHA, dehydroascorbate; normal, salt-free conditions; stressed, salt-treated conditions; H_2_O_2_, hydrogen peroxide; MDA, malondialdehyde; MDHAR, monodehydroascorbate reductase; APX, ascorbate peroxidase; GR, glutathione reductase.

**Figure 3 antioxidants-11-01077-f003:**
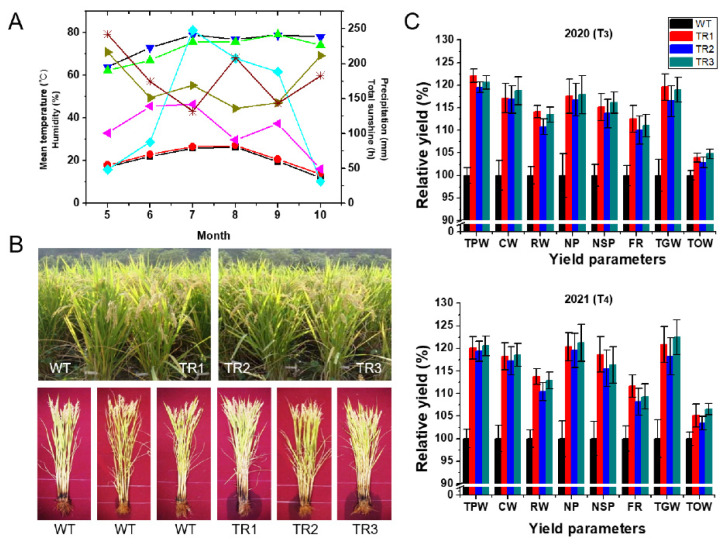
Agronomic traits of transgenic and wild-type rice plants grown under paddy-field conditions. (**A**) Weather conditions during the cultivation period (May to October) of 2020 and 2021, including mean temperature (2020, black square; 2021, red circle), precipitation (2020, cyan diamond; 2021, magenta left-triangle), total sunshine (2020, dark yellow right-triangle; 2021, wine star), and humidity (2020, green up-triangle; 2021, blue down-triangle). (**B**) Phenotypes of middle- (upper panel) and late-reproductive stage plants (lower panel) in 2021. (**C**) Relative agronomic traits of TR and WT plants in 2020 (T_3_) and 2021 (T_4_). TR, transgenic rice; WT, wild-type rice; TPW, total plant weight; CW, culm weight; RW, root weight; NP, number of panicles per hill; NSP, number of spikelets per panicle; FR, filling rate; TGW, total grain weight; TOW, 1000-grain weight.

**Figure 4 antioxidants-11-01077-f004:**
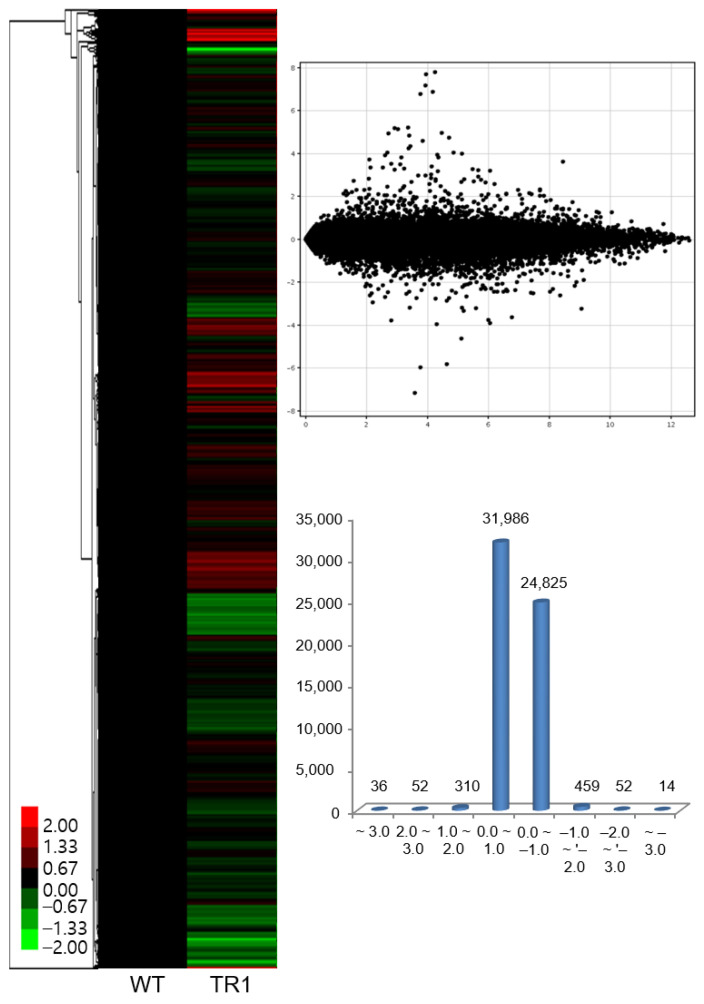
Microarray-based expression profiles of transgenic and wild-type rice plants. Microarray analysis was performed in order to determine the gene expression profiles of transgenic rice (TR) plants grown under the natural paddy fields. Heat map (**left**) and numbers (**right**) of up- and downregulated genes. WT, wild-type rice; TR, transgenic rice. Marked characteristics are annotated in Appendix A.

**Figure 5 antioxidants-11-01077-f005:**
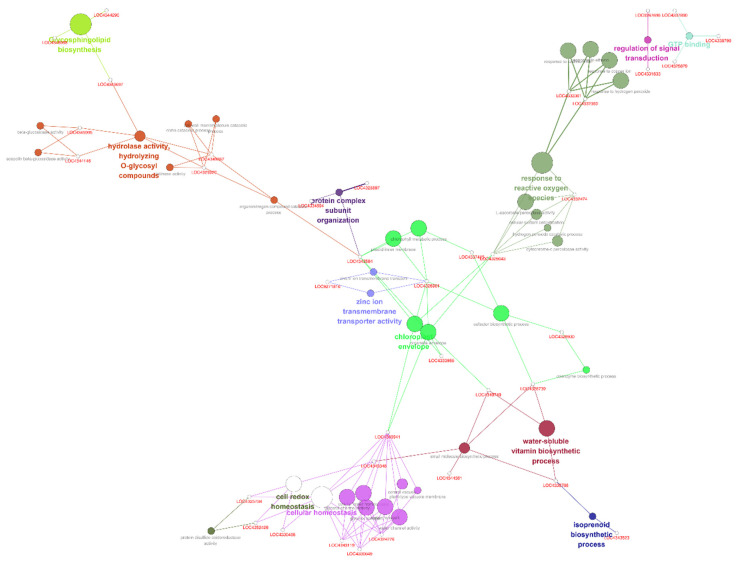
Genetic network of the genes upregulated in TR plants. Marked characteristics are annotated in Appendix A.

**Figure 6 antioxidants-11-01077-f006:**
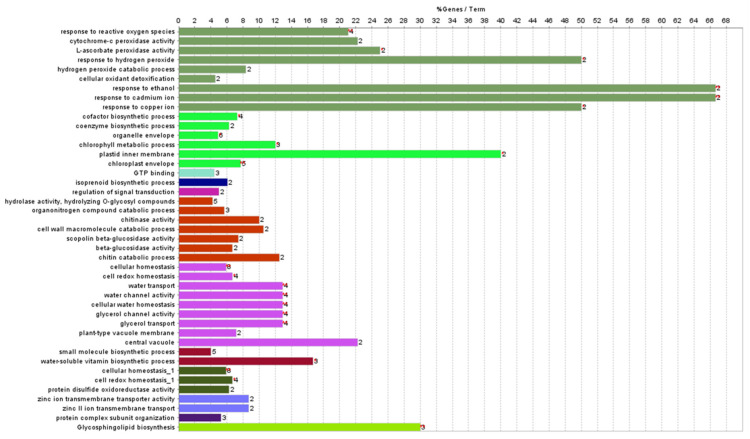
Gene ontology of the upregulated genes comprising genetic network in TR plants. Marked characteristics are annotated in Appendix A.

## Data Availability

Data is contained within the article.

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
