# Peer review of "Over-Expression of Dehydroascorbate Reductase Improves Salt Tolerance, Environmental Adaptability and Productivity in Oryza sativa"

_antioxidants, 2022, doi:10.3390/antiox11061077_

Round 1

Reviewer 1 Report

The presented manuscript reports the production and analysis of transgenic rice lines over-expressing a rice dehydroascorbate reductase gene under the control of a stress-inducible promoter from sweat potato. The manuscript is well written, the experimental data are well presented and they convincingly support the hypotheses.

I only have a few minor- or formal objections: 

Title:
As the manuscript is dealing with the ectopic expression of an extra copy of the OsDHAR1 gene under the control of a heterologous promoter, I suggest to change the title to 

"Over-expression of Dehydroascorbate Reductase Improves Salt Tolerance, Environmental Adaptability and Productivity in Oryza sativa" 
instead of the present title 

"High Expression of Dehydroascorbate Reductase Improves Salt Tolerance, Environmental Adaptability, and Productivity in Oryza sativa"

Abstract
Lines 43-44:
"dhar2-knockdown plants were susceptible to salt stress, and owing to poor seed maturation, exhibited reduced biomass (root growth) and grain yield ..."

Discussion
Page 16. Lines 51-52:
"In contrast, the DHAR2-knockdown transgenic lines, dhar2-1 and dhar2-2, which exhibited lower DHAR activity, were sensitive to salt ..."

(1)
dhr2 (DHR2) knockdown lines are mentioned in the Abstract and next in Discussion. However, such knockdown lines are not introduced or described in other parts of the manuscript. Please clarify this.
(2) Spelling: Which is the preferred spelling: dhr2 or DHR2? 
(Conventionally, lowercase italics is used for gene names, and full uppercase for protein names.)

"genetic mapping":
This term is repeatedly used at several parts of the manuscript. However, the term "genetic mapping" is conventionally used for linkage analysis of markers and traits, which usually results in genetic maps showing the linear order of genes along chromosomes.
The authors of the present manuscript use the term "genetic mapping" for either expression profiling or pathway analysis. Please correct and clarify at all occurrences. 

Author Response

Response to Reviewer 1 Comments

21 May 2022

Young-Saeng Kim, Ph.D.

Research Institute of Ulleung-do & Dok-do

Kyungpook National University

Daegu 41566, Republic of Korea

Phone: +82-53-951-7874

Fax: +82-53-953-3066

E-mail: kyslhh1228@hanmail.net

Dear Editor

Thank you for the critical comments and kind suggestions to improve our manuscript. We have carefully looked over and revised the manuscript according to all the comments of the Reviewers. The details we made in revised manuscript are listed below point by point. We wish you will find this manuscript now publishable in “Antioxidants”.

Point 1: Title

As the manuscript is dealing with the ectopic expression of an extra copy of the OsDHAR1 gene under the control of a heterologous promoter, I suggest to change the title to

"Over-expression of Dehydroascorbate Reductase Improves Salt Tolerance, Environmental Adaptability and Productivity in Oryza sativa"

instead of the present title

"High Expression of Dehydroascorbate Reductase Improves Salt Tolerance, Environmental Adaptability, and Productivity in Oryza sativa"

Response 1: Thank you for your kind suggestion to make the title of our manuscript clearer and more impact. According to the Reviewer1’s comment, we revised the title to "Over-expression of Dehydroascorbate Reductase Improves Salt Tolerance, Environmental Adaptability and Productivity in Oryza sativa"

Point 2: Abstract

Lines 43-44:

"dhar2-knockdown plants were susceptible to salt stress, and owing to poor seed maturation, exhibited reduced biomass (root growth) and grain yield ..."

Response 1: Thank you for your kind suggestion to make the dhar2-knockdown plants for our manuscript clearer. 43 lines of page 1.

Discussion

Page 16. Lines 51-52:

"In contrast, the DHAR2-knockdown transgenic lines, dhar2-1 and dhar2-2, which exhibited lower DHAR activity, were sensitive to salt ..."

Response 2: Thank you for your kind suggestion to make the dhar2-knockdown transgenic lines for our manuscript clearer. 7-8 lines of page 18. “the dhar2-knockdown transgenic lines, dhar2-1 and dhar2-2, which exhibited lower DHAR activity”

(1) dhr2 (DHR2) knockdown lines are mentioned in the Abstract and next in Discussion. However, such knockdown lines are not introduced or described in other parts of the manuscript. Please clarify this.

(2) Spelling: Which is the preferred spelling: dhr2 or DHR2?

(Conventionally, lowercase italics is used for gene names, and full uppercase for protein names.)

Response 3:

(1) At the request of the reviewers, we have inserted additional descriptions of the results for the dhar2-knockdown transgenic lines in 5-8 and 14-16 lines of page 10, 25-28 lines of page 11, and 35-44 lines of page 12.

(2) We revised DHAR2 to dhar2 in 7-8 lines of page 18 and 24-25 lines of page 22.

Point 3: "genetic mapping"

This term is repeatedly used at several parts of the manuscript. However, the term "genetic mapping" is conventionally used for linkage analysis of markers and traits, which usually results in genetic maps showing the linear order of genes along chromosomes.

The authors of the present manuscript use the term "genetic mapping" for either expression profiling or pathway analysis. Please correct and clarify at all occurrences.

Response 3:  Based on the reviewer’s suggestion, we corrected the manuscript. Please check that change from genetic mapping to genetic network in the manuscript.

Reviewer 2 Report

The article entitled “High Expression of Dehydroascorbate Reductase Improves Salt Tolerance, Environmental Adaptability, and Productivity in Oryza sativa” authored by Young-Saeng Kim et al deals with the identification of the target genes of stress-related transcription factors and putative cis-acting DNA elements in OsDHAR1-overexpressing plants under conditions of environmental stress using gene expression profiling and genetic mapping.

The work is clear and comprehensive. Information is presented by the authors in a well-structured and  detailed manner. Informative tables and figures and up to date cite references have been used. The conclusions are useful and are based on the experimental results.

The paper may be accepted in its present Form

Author Response

Response to Reviewer 2 Comments

21 May 2022

Young-Saeng Kim, Ph.D.

Research Institute of Ulleung-do & Dok-do

Kyungpook National University

Daegu 41566, Republic of Korea

Phone: +82-53-951-7874

Fax: +82-53-953-3066

E-mail: kyslhh1228@hanmail.net

Dear Editor

Thank you for the critical comments and kind suggestions to improve our manuscript. We have carefully looked over and revised the manuscript according to all the comments of the Reviewers. The details we made in revised manuscript are listed below point by point. We wish you will find this manuscript now publishable in “Antioxidants”.

Point 1:

The work is clear and comprehensive. Information is presented by the authors in a well-structured and detailed manner. Informative tables and figures and up to date cite references have been used. The conclusions are useful and are based on the experimental results.

The paper may be accepted in its present Form

Response 1: We thank you for your evaluation of our manuscript.

Reviewer 3 Report

I have carefully analyzed the paper and consider that from an agronomic point of view it is an important topic and has the potential of scientific contribution in this field.

The paper is well documented, and the research hypothesis is detailed enough at the end of the introductory part.

However, after the introductory part, the paper is difficult to understand. The research methods are not explained, the results are difficult to decipher, and the references are not written correctly. There are many mistakes and the paper does not have the necessary accuracy.

I recommend the following major (method) - minor improvements:
1. For the material and the method it is not enough to say that you worked according to the methods presented previously [25,26] - page 4, line 11-12; and on page 9, lines 16-18, list agronomic traits were scored, without presenting methods and references; I believe that the whole chapter of the material and the method must be corrected and all the methods and references used must be added, in order to better understand the experiment.

2. The figures presented in the results are not eligible, they must be corrected in order to be legible.
3. References must be written in accordance with the drafting instructions.

Author Response

Response to Reviewer 3 Comments

21 May 2022

Young-Saeng Kim, Ph.D.

Research Institute of Ulleung-do & Dok-do

Kyungpook National University

Daegu 41566, Republic of Korea

Phone: +82-53-951-7874

Fax: +82-53-953-3066

E-mail: kyslhh1228@hanmail.net

Dear Editor

Thank you for the critical comments and kind suggestions to improve our manuscript. We have carefully looked over and revised the manuscript according to all the comments of the Reviewers. The details we made in revised manuscript are listed below point by point. We wish you will find this manuscript now publishable in “Antioxidants”.

Point 1:

  1. For the material and the method it is not enough to say that you worked according to the methods presented previously [25,26] - page 4, line 11-12; and on page 9, lines 16-18, list agronomic traits were scored, without presenting methods and references; I believe that the whole chapter of the material and the method must be corrected and all the methods and references used must be added, in order to better understand the experiment.

Response 1:

(1) [25,26] - page 4, line 11-12: We replaced references [25,26] in line 11 of page 4 with the following references.

  1. Hiei, Y.; Ohta, S.; Komari, T.; Kumashiro, T., Efficient transformation of rice (Oryza sativa L.) mediated by Agrobacterium and sequence analysis of the boundaries of the T-DNA. Plant J 1994, 6, 271-82.
  2. Park, S. I.; Kim, J. J.; Shin, S. Y.; Kim, Y. S.; Yoon, H. S., ASR enhances environmental stress tolerance and improves grain yield by modulating stomatal closure in rice. Front Plant Sci 2020, 10, 1752.

(2) on page 9, lines 16-18: At the request of reviewers, the method for secoring agronomic traits was added as follows in lines 8-18 of page 9.

When the plants reached maturity and the grains ripened, yield components were scored by digging up 8 rice plants per plot to a radius of 15 cm. After washing the plant roots, total plant weight (g), stem weight (g), and root weight were measured and number of panicles per hill and number of spikelets per panicle were counted. The filling rate (%) of the rice plants was calculated relative to 100% of WT rice plants, which were defined had expressed the degree of the filling rate as panicles and spikelets under field conditions. To measure the total grain weight (g), and 1000-grain weight (g), whole seeds per plant were collected and weighed, and then 1000 seeds were counted and weighed additionally.

Point 2: The figures presented in the results are not eligible, they must be corrected in order to be legible

Response 2:  Following the reviewer’s suggestion, the manuscript was corrected in order to be legible for the figures presented in the results.

Point 3: References must be written in accordance with the drafting instructions.

Response 3: As requested, we corrected the reference according to the draft guidelines.
